# FedQV: Leveraging Quadratic Voting in Federated Learning

## Abstract

Federated Learning (FL) permits different parties to collaboratively train a global model without disclosing their respective local labels. A crucial step of FL, that of aggregating local models to produce the global one, shares many similarities with public decision-making, and elections in particular. In that context, a major weakness of FL, namely its vulnerability to poisoning attacks, can be interpreted as a consequence of the *one person one vote* (henceforth *1p1v*) principle underpinning most contemporary aggregation rules. In this paper, we propose FEDQV, a novel aggregation algorithm built upon the *quadratic voting* scheme, recently proposed as a better alternative to *1p1v*-based elections. Our theoretical analysis establishes that FEDQV is a truthful mechanism in which bidding according to one's true valuation is a dominant strategy that achieves a convergence rate that matches those of state-of-the-art methods. Furthermore, our empirical analysis using multiple real-world datasets validates the superior performance of FEDQV against poisoning attacks. It also shows that combining FEDQV with unequal voting "budgets" according to a reputation score increases its performance benefits even further. Finally, we show that FEDQV can be easily combined with Byzantine-robust privacy-preserving mechanisms to enhance its robustness against both poisoning and privacy attacks.

## 1 Introduction

Federated Learning (FL) has emerged as a promising privacy-preserving paradigm for conducting distributed collaborative model training across parties that do not want to disclose their local data. Agreeing on a common global model in Federated Learning shares many similarities with public decision-making and elections in particular. Indeed, the weights of local model updates of a party (client) can be seen as votes of preference that affect the global model resulting from an aggregation rule applied at the centralised server of an FL group. FEDAVG McMahan et al. (2018) has been the "de facto" aggregation rule used in FL tasks such as Google's emoji and next-word prediction for mobile device keyboards Ramaswamy et al. (2019); Hard et al. (2018). In FEDAVG the global model is produced from a simple weighted averaging of local updates with weights that represent the amount of data that each party has used for its training.

**The problem**  Recent work Blanchard et al. (2017) has shown that FEDAVG is vulnerable to poisoning attacks, as even a single attacker can degrade the global model by sharing faulty local updates of sufficiently large weight. Such attacks become possible because FEDAVG treats all local data points equally. In essence, the aggregation rule, when seen at the granularity of individual training data, resembles the *one person one vote (1p1v)* election rule of modern democratic elections. In this context, the server distributes votes (weights) to a party in accordance with the amount of its training data, which may be regarded as its population. This, however, may confer an unjust advantage to malicious parties with large training datasets.

**Our approach**  To address this issue, we propose a novel aggregation rule inspired by elections based on *Quadratic Voting* Lalley & Weyl (2018) (henceforth QV). In QV, each party is given a voting budget that can be spent on different rounds of voting. Within a particular vote, an individual has to decide the number of "credit voices" to commit, whose square root is what impacts the corresponding outcome of the vote. QV has been proposed as a means to break out from the tyranny-of-the-majority vs. subsidising-the-minority dilemma of election systems Posner & Weyl (2015).

Its formal analysis Weyl (2017) under a game theoretic price-taking model, has shown that QV outperforms *1p1v* in terms of efficiency and robustness. Importantly, it has the unique capacity to deter collusion attacks by effectively outlawing extreme behaviours.

**Our contributions.** In this paper, we propose FEDQV, a novel FL aggregation scheme that draws inspiration from quadratic voting. Our objective is to mitigate the ability of malicious peers who may have, or falsely claim to have, large datasets, to impose a disproportional damage on the global model – a vulnerability inherent in the FEDAVG that applies the *1p1v* principle at the granularity of individual votes. First, we demonstrate that the incorporation of QV into the FL setting restricts the ability of malicious peers to inflict high damages by taxing their credit voices more than linear. Figure 1 illustrates a toy use case with two benign and one malicious party engaged in a poisoning attack, with the dataset sizes set to $\{1, 1, 2\}$, respectively. In contrast to FEDAVG, which allocates aggregation weights as $\{1, 1, 2\}$, QV allocates weights as $\{1, 1, \sqrt{2}\}$, effectively limiting the malicious party's influence.

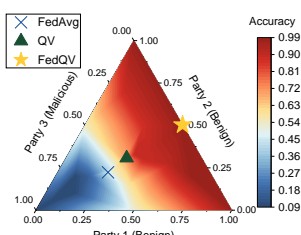

Figure 1: Global model weights (position within the triangle) and corresponding test accuracy (color-coded) with three parties (two benign and one malicious). FE-DAVG is located at the bottom left corner ; QV is positioned around the centre ; FEDQV is situated along the right triangle side. Details of the experimental setup are provided in Appendix C.4.

To capture each party's preference for voting and enhance the detection of malicious updates, we require parties to submit the similarity of their local model with the previous round global model as their aggregation weight. Furthermore, in response to potential malicious attempts, we also introduce a truthfulness mechanism, FEDQV, to our application of QV. This mechanism employs a masked voting rule and a limited budget to hide the vote calculation process from parties, preventing them from knowing the exact votes they have cast. These measures act as a deterrent against parties providing false information to evade penalties, which may exclude them from the current and following rounds. Returning to our previous toy example, FEDQV results in the allocation of weights $\{1, 1, 0\}$, as illustrated in Figure 1, that effectively excludes the malicious party from the aggregation, thereby increasing the accuracy of the resulting global model.

In election-related applications, QV allocates equal budgets to all voters, reflecting the democratic principle of equal rights. However, in our adaptation of QV for FL, it makes sense to allocate more votes to benign peers and limit the influence of malicious ones. We achieve this by employing unequal budgets, which are tied to a reputation score for each peer, as discussed in Section 5.6. Furthermore, we design FEDQV such that it can be easily combined with existing privacy-guaranteed mechanisms to thwart *inference and reconstruction attacks* Melis et al. (2019); Zhu et al. (2019).

In terms of theoretical contributions, we present an extensive analysis in order to: 1) establish convergence guarantees, and 2) prove the truthfulness of our method. We also conduct a thorough experimental evaluation for studying the accuracy, convergence, and resilience of our proposed mechanism against state-of-the-art Byzantine attacks on multiple benchmark datasets.

Our final contribution lies in extending the versatility of our core FEDQV scheme, by facilitating its seamless integration with state-of-the-art Byzantine-robust FL defences. This enables FEDQV to serve as a complementary component, ultimately boosting the robustness of these existing defences, rather than being seen as a competitor. Notably, implementing these defences atop FEDQV consistently yields superior results compared to employing them on top of FEDAVG.

**Our findings.** Using a combined theoretical and experimental evaluation, we show that:

• FEDQV is a truthful mechanism and is theoretically and empirically compatible with FEDAVG in terms of accuracy and convergence under attack and no-attack scenarios.

• FEDQV consistently outperforms FEDAVG under various SOTA poisoning attacks, especially for local model poisoning attacks improving the robustness to such attacks by a factor of at least $4\times$.

• The combination of FEDQV with a reputation model to assign unequal credit voice budgets to parties according to their respective reputations, improves robustness against poisoning attacks by at least 26% compared to the baseline FEDQV that uses equal budgets.

• We show that integrating FEDQV with established Byzantine-robust FL defences, including Multi-Krum Blanchard et al. (2017), Trimmed-Mean Yin et al. (2018), and Reputation Chu et al. (2022), results in substantial enhancements in accuracy and reductions in the attack success rate under state-of-the-art attacks when compared to the original defence methods.

## 2 RELATED WORK

### 2.1 ELECTION MECHANISMS IN FL

Election mechanisms are widely used in distributed systems for choosing a coordinator from a collection of processes Garcia-Molina (1982); Alford et al. (1985). Likewise, there exist works that explore the value of the election mechanism for the aggregation step of FL. Plurality voting is employed in *FedVote* Yue et al. (2022) and *FedVoting* Liu et al. (2021) for treating the validation results as votes to decide the optimal model. Also in Sohn et al. (2020), the authors propose two forms of election coding for discovering majority opinions for the aggregation step. *DETOX* Rajput et al. (2019) proposes a hierarchical aggregation step based on majority votes upon groups of updates. Finally, *DRACO* Chen et al. (2018) and *ByzShield* Konstantinidis & Ramamoorthy (2021) also employ majority voting to fend off attacks against the aggregation step. All the aforementioned election mechanisms suffer from the tyranny of the majority problem in election systems Sartori (1987). In FL, this means that if attackers manage to control the majority of votes, then via poisoning their tyranny will manifest itself as a degradation of the accuracy of the FL model used by the minority.

To address these limitations, QV is proposed as a solution that combines simplicity, practicality, and efficiency under relatively broad conditions. QV considers a quadratic vote pricing rule, inspired by economic theory, under which voters can purchase votes at ever-increasing prices within a predetermined voting budget. The advantages of QV over *1p1v* have a rigorous theoretical basis, which of course applies also to the use of QV in FL. For any type of symmetric Bayes-Nash equilibrium, the price-taking assumption approximately holds for all voters, as a result, the expected inefficiency of QV is bounded by constant Lalley et al. (2016). This theoretical analysis Chandar & Weyl (2019); Tideman & Plassmann (2017) combined with strong empirical validation, both at the laboratory Casella & Sanchez (2019) and on the field Quarfoot et al. (2017), suggest that QV is near-perfectly efficient and more robust than *1p1v* which, as already explained, forms the basis of contemporary FL aggregation mechanisms. The advantages of QV can also be observed from the viewpoint of collusion, which is generally deterred either by unilateral deviation incentives or by the reactions of non-participants Weyl (2017).

### 2.2 BYZANTINE-ROBUST FL AGGREGATION AGAINST PRIVACY ATTACKS

There exist several Byzantine-robust FL aggregation methods for mitigating Byzantine attacks either by leveraging statistic-based outlier detection techniques Blanchard et al. (2017); Yin et al. (2018); Xie et al. (2019); Chu et al. (2022) or by utilising auxiliary labelled data collected by the aggregation server in order to verify the correctness of the received gradients Guo et al. (2021); Cao et al. (2021). Both approaches, though, require examining the properties of the updates of individual parties, which can jeopardise their privacy due to inference Melis et al. (2019) and reconstruction attacks Zhu et al. (2019); Geiping et al. (2020) mounted by an honest but curious aggregation server. Contrary to those approaches, in FEDQV the analysis of local updates and the calculation of corresponding weights is done locally at the peers using provably truthful mechanisms. This allows for the implementation of FEDQV effectively using cryptographic techniques, such as *the secure aggregation scheme* Bonawitz et al. (2016) and *Fully Homomorphic Encryption* Aono et al. (2017), without being exposed to inference and reconstruction attacks at the aggregation server. It is worth noting that while there are alternative privacy-guaranteed mechanisms available in FL, such as differential privacy Dwork (2006); Du et al. (2020) and model inversion Zhao et al. (2022), they do not provide the same level of security as cryptology-based methods Zhu et al. (2019). However, it is important to acknowledge that cryptographic methods are typically suitable for simple and specific computations like weighted averaging in FEDAVG and FEDQV. Hence, these methods are not applicable to more complex computations and data analyses required for the most Byzantine-robust FL aggregations. Although a few other FL aggregation approaches Ma et al. (2022); So et al. (2020) can be adapted to incorporate cryptographic techniques, they still rely on majority voting as the aggregation scheme, which can be seamlessly integrated with FEDQV to enhance its robustness against Byzantine attacks.

## 3 METHODOLOGY

### 3.1 FEDERATED LEARNING SETTING

Consider an FL system involving $N$ parties and a central server. During training round $t$, a subset of parties $\mathcal{S}^t$ is selected to participate in the training task. Party $i$ has the local dataset $\mathcal{D}_i$ with $|\mathcal{D}_i|$ samples (voters), drawn from non-independent and non-identically (Non-IID) distribution $\mathcal{X}_i(\mu_i, \sigma_i^2)$. The goal of using FL is to learn a global model for the server. Given the loss function $\ell(\boldsymbol{w}; \mathcal{D})$, the objective function of FL can be described as $\mathcal{L}(\boldsymbol{w}) = \mathbb{E}_{\mathcal{D} \sim \mathcal{X}}[\ell(\boldsymbol{w}; \mathcal{D})]$. Therefore, the task becomes: $\boldsymbol{w}^* = \arg\min_{\boldsymbol{w} \in \mathbb{R}^d} \mathcal{L}(\boldsymbol{w})$. To find the optimal $\boldsymbol{w}^*$, Stochastic Gradient Descent (SGD) is employed to optimise the objective function. Let $T$ be the total number of every part's SGD, $E$ be the local iterations between two communication rounds, and thus $\frac{T}{E}$ is the number of communication rounds.

The FL model training process entails several rounds of communication between the parties and the server, including broadcasting, local training, and aggregation, as demonstrated in Algorithm 1. For aggregation rule, FEDAVG uses the fraction of the local training sample size of each party over the total training samples as the weight of a party: $\boldsymbol{w}^{t+1} = \sum_{i \in \mathcal{S}^t} |\mathcal{D}_i| \cdot \boldsymbol{w}_i^t / \bigcup_{i \in \mathcal{S}^t} |\mathcal{D}_i|$. Similar to *1p1v*, each sample here represents a single voter, and since party $i$ possesses $|\mathcal{D}_i|$ samples, it is able to cast $|\mathcal{D}_i|$ votes for its local model during the aggregation. Hence, the global proposal is a combination of all parties' local proposals weighted by their votes.

### 3.2 FEDQV: QUADRATIC VOTING IN FL

We use QV in FL to overcome the drawback of *1p1v*, which improves the robustness of aggregation in comparison to FEDAVG without compromising any efficiency. Our QV-based aggregation algorithm consists of two key components: *similarity computation* and *voting scheme*.

**Similarity Computation (Party Side):** In round $t$, based on the server instructions, party $i$ ($i \in \mathcal{S}^t$) trains its local model $\boldsymbol{w}_i^t$ which can be regarded as its local proposal. Then the party calculates its similarity score $s_i^t$ between the previous global model $\boldsymbol{w}^{t-1}$ and its local model using the cosine similarity function. Notably, the cosine similarity function can be adapted to different similarity metrics, such as L2 distance, to better suit specific tasks. In this context, a higher $s_i^t$ value indicates a stronger agreement with the previous global model(proposal). Once selected parties finish training, they send their updates $\boldsymbol{w}i^t$ to the server, with the message containing $s_i^t$ and $\mathcal{D}_i$.

**Voting Scheme (Server Side):** Upon receiving the updates and similarity scores from selected parties, the server proceeds with the following steps:

(i) The server normalises the similarity scores $s_i^t$ using Min-Max Scaling to obtain $\bar{s}_i^t \in [0, 1]$;

(ii) The server penalises parties with abnormal similarity scores ($\bar{s}_i^t \leq \theta$ or $\bar{s}_i^t \geq 1 - \theta$), where $\theta$ is the similarity threshold. This addresses excessively large or small similarity scores, which are considered suspicious. Penalties are applied by adjusting their budget $B_i$ with the formula: $B_i = \max\left(0, B_i + \ln \bar{s}_i^t - 1\right)$;

(iii) The server calculates the voice credit $c_i^t$ for party $i$ utilising the masked voting rule $\mathcal{H}$:

$$c_i^t = \mathcal{H}(\bar{s}_i^t) = \left(-\ln \bar{s}_i^t + 1\right) \mathbb{1}_{\theta < \bar{s}_i^t < 1-\theta} \tag{1}$$

Here, the voice credit signifies the price party $i$ is required to pay in round $t$ for its local proposal. Parties with higher similarity scores, indicating stronger agreement with the global proposal, receive fewer credit votes from the server.

(iv) The server checks the budget $B_i$ for each party $i$ and employs QV to compute their final votes $v_i^t$:

$$v_i^t = \sqrt{\min\left(|\mathcal{D}_i| c_i^t, \max\left(0, B_i\right)\right)} \tag{2}$$

Subsequently, the server updates the budget as follows: $B_i = \max(0, B_i - (v_i^t)^2)$.

Thus, the server determines the weight $(v_i^t)$ of party $i$ for aggregation and generates the updated global model $\boldsymbol{w}^{t-1}$. Algorithm 1 summarises all these steps of FEDQV.

Importantly, due to the masked voting rule $\mathcal{H}$, only the server possesses knowledge of each party's remaining budget and the number of actual votes cast in the current round. This feature ensures that

---

**Algorithm 1:** FEDQV

---

**Input** : $\boldsymbol{w}^0 \leftarrow$ random initialisation; $B, \theta \leftarrow$ FEDQV parameters

**Server :**

1 **for** *Iteration* $t \leftarrow 1$ **to** $\frac{T}{E}$ **do**
2      Broadcast $\boldsymbol{w}^{t-1}$ to randomly selected set of parties $\mathcal{S}^t$ ($|\mathcal{S}^t| = \mathcal{C} \geq 1$);
3      Receive *the local updates* $(\boldsymbol{w}^t, s^t, |\mathcal{D}|)$ and compute the normalised $\bar{s}_i^t$ ;
4      **for** $i \leftarrow 1$ **to** $N$ **do in parallel**
5          **if** $\bar{s}_i^t \leq \theta$ *or* $\bar{s}_i^t \geq 1 - \theta$ **then**
6             |   Update $B_i \leftarrow \max\left(0, B_i + \ln \bar{s}_i^t - 1\right)$
7          *Credit voice* $c_i^t \leftarrow$ Equation 1 and *Vote* $v_i^t \leftarrow$ Equation 2;
8          *Budget* $B_i \leftarrow \max\left(0, B_i - (v_i^t)^2\right)$
9      **return** $\boldsymbol{w}_n^t \leftarrow \sum_{i=1}^{N} \frac{v_i^t}{\sum_{i=1}^{M} v_i^t} \boldsymbol{w}_{i,n}^t$

**Party :**

1 **for** *Party* $i \in \mathcal{S}^t$ **do in parallel**
2      Receive *the global update* $\boldsymbol{w}^{t-1}$ and conduct local training $\boldsymbol{w}_i^t \leftarrow \boldsymbol{w}_i^{t-1} - r_{t-1} \frac{\partial \ell_i(\boldsymbol{w}^{t-1}; \mathcal{D}_i)}{\partial \boldsymbol{w}}$;
3      Calculate *the similarity score* $s_i^t \leftarrow \frac{\langle \boldsymbol{w}_i^t, \boldsymbol{w}^{t-1} \rangle}{\|\boldsymbol{w}_i^t\| \cdot \|\boldsymbol{w}^{t-1}\|}$ and send back $(\boldsymbol{w}_i^t, s_i^t, |\mathcal{D}_i|)$

---

parties remain unaware of the inner workings of the credit voice allocation process for aggregation. Consequently, even if parties possess a comprehensive understanding of how FEDQV functions on the server side, they remain incapable of strategizing or predicting their credit voice allocation. In cases where parties attempt to manipulate the similarity scores, which are relatively easy to detect as abnormal and falling beyond the threshold, the server takes punitive actions. These actions include resetting their voice credits to zero, excluding them from the aggregation process, and reducing their remaining budgets, rendering them incapable of causing harm to the global model.

**Benefits of FEDQV 1. Truthful Mechanism.** FEDQV is a *truthful* mechanism Blumrosen & Nisan (2007) as we prove in Theorem 4.5. This means that this mechanism compels the parties, even malicious ones, to tell the truth about their votes (weights) for aggregation, rather than any possible lie. This truthfulness is reinforced by several key factors: *(i) Punitive Measures*: FEDQV, with its masked voting rule and limited budget, has provisions to penalise and remove malicious participants, acting as a strong deterrent against manipulation attempts; *(ii) Limited Influence*: Even if a manipulated similarity score is accepted, the influence the malicious participant can exert is inherently constrained due to the nature of QV, minimising the potential damage.

**2. Ease of Integration and Compatibility.** FEDQV is highly adaptable and can be seamlessly integrated into Byzantine-robust FL defence schemes with minimal adjustments, specifically by modifying the aggregation weight calculation while leaving other algorithm components unchanged. This integration is demonstrated in Section 5.7. Furthermore, similar to FEDAVG, FEDQV boasts efficient communication and simplicity, rendering it compatible with various mechanisms employed in FL. It can effortlessly incorporate the regularisation, sparsification, and privacy modules, encompassing techniques such as clipping McMahan et al. (2018), gradient compression Sattler et al. (2019), differential privacy Dwork (2006), and secure aggregation Bonawitz et al. (2016).

### 3.3 FEDQV WITH ADAPTIVE BUDGETS

In democratic elections, all individuals are typically granted equal voting rights, entailing an equal voting budget. In FL, however, it often makes sense to give malicious parties fewer votes than honest ones. Thus to improve the robustness of standard FEDQV, we combine it with the reputation model in Chu et al. (2022) to assign an unequal budget based on the reputation score of parties in each round $t$. Specifically, if a party's reputation score $R^t$ surpasses a predefined threshold $\lambda$, we increase their budget, and vice versa. We present a summary of this combination in Algorithm 2, with a detailed explanation provided in Appendix B, expanding on the well-established components from the

original paper. We provide empirical evidence in section 5 showcasing the substantial performance improvements achieved by the enhanced version of FEDQV featuring an adaptive budget.

---

**Algorithm 2:** FedQV with Adaptive Budget

**Input** : $w_i^t, c_i^t, B_i^t \leftarrow$ FEDQV; $\kappa$,$a$,$W$,$M$,$\lambda$,$\delta \leftarrow$ *Reputation model* parameters

1 **for** $i \in \mathcal{S}^t$ **do**
2     **for** $j \leftarrow 1$ **to** $M$ **do**
3        *Subjective Observations* $(P_i^t, N_i^t) :=$ IRLS $(w_{i,j}^t, \delta)$;
4     *Reputation Score* $R_i^t :=$ Rep $(P_i^t, N_i^t, \kappa, a, W)$
5     *Budget* $B_i^t \leftarrow R_i^t \mathbb{1}_{\lambda \leq R_i^t} + B_i^t$, *Credit voice* $c_i^t \leftarrow (R_i^t + c_i^t) \mathbb{1}_{\lambda \leq R_i^t}$

---

## 4 THEORETICAL ANALYSIS

In this section, we show that the convergence for FEDQV is guaranteed in bounded time and that FEDQV is a truthful mechanism. Our first major result is Theorem 4.1 that states FEDQV converges to the global optimal solution at a rate of $\mathcal{O}(\frac{1}{T})$, where $T$ is the total number of SGD, for strongly convex and smooth functions with non-iid data. Regarding the performance of our algorithm in terms of metric average accuracy and convergence as will be illustrated in the following section, we show that it is consistent with our theoretical analysis. Our second major result is Theorem 4.5 which states that FEDQV is a truthful mechanism. Fully detailed proofs are provided in the Appendix A.

### 4.1 CONVERGENCE

Suppose the percentage of attackers in the whole parties is $m$, we denote

$$\mathcal{M}_i(\boldsymbol{w}_i^t) = \begin{cases} * & \text{if } i \in \text{malicious parties} \\ \nabla \ell(\boldsymbol{w}_i^t; \mathcal{D}_i^t) & \text{if } i \in \text{honest parties} \end{cases}$$

Where $*$ stands for an arbitrary value from the malicious parties. Under four mild and standard assumptions for such types of analysis in accordance with recent works Yin et al. (2018); Xie et al. (2019); Yu et al. (2019); Cao et al. (2021); Chu et al. (2022); Cao et al. (2023), along with the support of Lemmas outlined in the Appendix A.2.1, we have

**Theorem 4.1.** *Under Assumptions A.1, A.2, A.3 and A.4, Choose* $\alpha = \frac{L+\mu}{\mu L}$ *and* $\beta = 2\frac{(L+1)(L+\mu)}{\mu L}$, *then* FEDQV *satisfies*

$$\mathbb{E}\,\mathcal{L}(\boldsymbol{w}^T) - \mathcal{L}(\boldsymbol{w}^*) \leq \frac{L + 2Lr_{T-1}\varpi}{2\varphi + T}\left(\varphi\,\mathbb{E}\left\|\boldsymbol{w}^0 - \boldsymbol{w}^*\right\|_2^2 + \frac{\alpha^2}{2}\Delta\right) + \frac{L\varpi^2}{2} \tag{3}$$

*Where*

$$\Delta = (E-1)^2\,\mathcal{G}_{\boldsymbol{w}}^2 + (1-2\theta)\,\mathcal{CV}_{\boldsymbol{w}}\sqrt{B}, \ \varphi = \alpha(L+1), \ \varpi = mN\mathcal{G}_{\boldsymbol{w}}r_{T-1}\sqrt{4+6\theta-\theta^2}$$

*Remark* 4.2. According to Theorem 4.1 and Theorem A.9, FEDQV obtains a convergence rate of $\mathcal{O}(\frac{1}{T})$ irrespective of the presence or absence of adversarial participants, which is comparable to the convergence rate of FEDAVG Li et al. (2020).

*Remark* 4.3. The error rate exhibits dependence on the budget $B$, the similarity threshold $\theta$, and the percentage of malicious parties $m$. It is noteworthy that a larger budget allocation, a reduction in the similarity threshold, or an augmentation in the proportion of malicious parties induce more pronounced disparities in model updates, consequently resulting in an elevated error rate. The impact of these hyperparameters is shown in Figure 5 in the Appendix C.6.

### 4.2 TRUTHFULNESS

The FEDQV mechanism belongs to a single-parameter domain since the single real parameter votes $v_i$ directly determines whether party $i$ will be able to join the aggregation. In addition, it is normalised according to the definition in the game theory Blumrosen & Nisan (2007) that for every $v_i$, $v_{-i}$ such that $f(v_i, v_{-i}) \notin W_i$, $p_i(v_i, v_{-i}) = 0$. Here, $v_{-i}$ denotes the votes cast by all other parties except for $i$, $W_i$ represents the subset of participants in aggregation, $f$ is the outcome of the voting scheme,

and $p_i$ is the payment function that $p_i(v_i, v_{-i}) = v_i^2$ in FEDQV. The following is the definition of truthfulness and lemmas that we use in the proof of the Theorem 4.5 in accordance with monotone and critical value in the game theory Blumrosen & Nisan (2007).

**Definition 4.4.** A mechanism $(f, p_1, ..., p_n)$ is called truthfulness if for every party $i$, we denote $a = f(v_i, v_{-i})$ and $a' = f(v_i', v_{-i})$, then $v_i(a) - p_i(v_i, v_{-i}) \geq v_i(a') - p_i(v_i', v_{-i})$.

Intuitively this means that party $i$ with $v_i$ would prefer "telling the truth" $v_i$ to the server rather than any possible "lie" $v_i'$ since this gives him higher (in the weak sense) utility.

Based on Lemma A.10 and A.11 in Appendix A.3.1, we have:

**Theorem 4.5.** FEDQV *is incentive compatible (truthful).*

*Remark* 4.6. Regarding the concept of truthfulness, it theoretically ensures that being honest is the dominant strategy since providing manipulated similarity scores may lead to penalties and removal from the system due to the masked voting rule $\mathcal{H}$ and limited budget $B$. This is an integral part of the nature of QV embedded within our FEDQV framework.

## 5 EXPERIMENTS

### 5.1 EXPERIMENTAL SETTING

**Datasets and global models**   We implement the typical FL setting where each party owns its local data and transmits/receives information to/from the central server. To demonstrate the generality of our method, we train different global models on different datasets. We use four popular benchmark datasets: MNIST LeCun (1998), Fashion-MNIST Xiao et al. (2017), FEMNIST Caldas et al. (2019) and CIFAR10 Krizhevsky (2009). We consider a multi-layer CNN same as in McMahan et al. (2018) for MNIST, Fashion-MNIST and FEMNIST, and the ResNet18 He et al. (2016) for CIFAR10.

**Non-IID setting**   In order to fulfil the setting of a heterogeneous and unbalanced dataset for FL, we sample from a Dirichlet distribution with the concentration parameter $\iota = 0.9$ as the Non-IID degree as in Bagdasaryan et al. (2020); Hsu et al. (2019), with the intention of generating non-IID and unbalanced data partitions. Moreover, we have examined the performance across varying levels of non-IID data, spanning from 0.1 to 0.9, as depicted in Appendix C.5.

**Parameter Settings**   The server selects 10 ($\mathcal{C}$) out of 100 ($N$) parties to participate in each communication round and train the global models for 100 communication rounds($\frac{T}{E}$). We set the model hyper-parameters budget $B$ and the similarity threshold $\theta$ to 30 and 0.2 respectively based on the hyper-parameter searching. All additional settings are provided in the Appendix C.1.

### 5.2 EVALUATED POISONING ATTACKS

Our paper addresses two distinct attack schemes:

• **Data poisoning**: Attackers submit the true similarity score based on their poisoned updates, including **Labelflip Attack** Fang et al. (2020), **Gaussian Attack** Zhao et al. (2022), **Backdoor** Gu et al. (2019), **Scaling Attack** Bagdasaryan et al. (2020), **Neurotoxin** Zhang et al. (2022).

• **Model poisoning**: Attackers submit the true similarity score based on their clean updates and poison their model, including: **Krum Attack** Fang et al. (2020), **Trim Attack** Fang et al. (2020), and Aggregation-agnostic attacks: **Min-Max** and **Min-Sum** Shejwalkar & Houmansadr (2021)

Moreover, we introduce an adaptive attack, **QV-Adaptive**, tailored for FEDQV, leveraging the AGR-agnostic optimisations Shejwalkar & Houmansadr (2021) within the LMP framework Fang et al. (2020) to manipulate both the similarity score and the local model.

The details of these attacks are in Appendix C.3. It is noteworthy that Labelflip, Gaussian, Krum, Trim, Min-Max, Min-Sum and QV Adaptive attacks are untargeted attacks, whereas, Backdoor, Scaling and Neurotoxin attacks are targeted attacks. We confine our analysis to the worst-case scenario in which the attackers submit the poisoned updates in every round of the training process for all attack strategies with the exception of the Scaling attack.

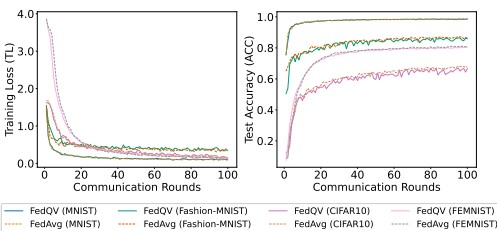

(a) Training Loss and ACC for 100 epochs of FEDQV and FEDAVG in four benchmark datasets under no attack scenario.

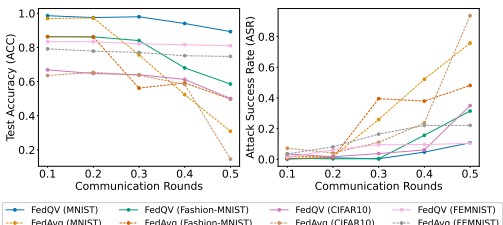

(b) ACC and ASR for 100 epochs of FEDAVG, FEDQV in four benchmark datasets under Backdoor attack with varying $m$ from 10% to 50%.

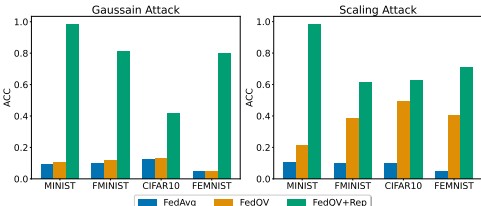

(c) ACC for 100 epochs of FEDAVG, FEDQV, FEDQV + REP(FEDQV with reputation model) in four benchmark datasets under 2 attack scenarios with 50% malicious parties.

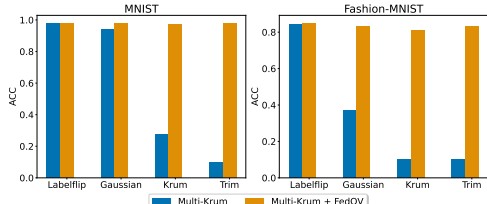

(d) Average test accuracy for 100 epochs of Krum and Multi-Krum + FEDQV on two benchmark datasets under 4 untargeted attack scenarios with 30% malicious parties.

## 5.3 PERFORMANCE METRICS

We use the average test accuracy (ACC) of the global model to evaluate the result of the aggregation defence for poisoning attacks. In addition, there are targeted attacks that aim to attack a specific label while keeping the accuracy of classification on other labels unaltered. Therefore, besides ACC, we choose the attack success rate (ASR) to measure how many of the samples that are attacked, are classified as the target label chosen by malicious parties.

## 5.4 CONVERGENCE

We evaluate the convergence of FEDAVG and FEDQV in the aforementioned four datasets without attack. We plot the training loss and ACC of the global models trained via FEDQV and FEDAVG in Figure 2a. We observe that, in the absence of Byzantine attacks, the global model trained using FEDQV converges as fast as that under FEDAVG for all four datasets, aligning with Theorem 4.1.

## 5.5 DEFENCE AGAINST POISONING ATTACKS

We present ACC and ASR results of global models trained using both FEDAVG and FEDQV under the 10 aforementioned attacks, with 30% malicious parties for all four datasets, in Table 1. In data poisoning attacks, the results consistently demonstrate that FEDQV outperforms FEDAVG, achieving the highest ACC with the smallest standard error. When considering targeted attacks, FEDQV again stands out, displaying the highest ACC along with the lowest ASR when compared to FEDAVG.In the context model poisoning attacks, FEDQV consistently outperforms FEDAVG, except for the QV-Adaptive attack, which is tailored for FEDQV. Especially for local model poisoning attacks: Trim and Krum attacks, FEDQV outperforms FEDAVG by at least *4 times* in terms of accuracy.

Then we vary the percentage of attackers from 10% to 50% in Figure 2b under the backdoor attack. Remarkably, FEDQV outperforms the baseline regarding ACC and ASR across all scenarios, even when half the parties are malicious. To investigate the behaviour of FEDQV in scenarios with finer gradations, we also evaluate it with small, realistic percentages of attackers, same as in Shejwalkar et al. (2022), in Table 2 and Appendix 6. However, we notice that none of these methods yields satisfactory accuracy results for Gaussian and Scaling attacks. To address this, we present the enhanced version of FEDQV with an adaptive budget assigned according to a reputation model.

## 5.6 ADAPTIVE BUDGET

The performance of FEDQV with an adaptive budget for the three evaluated methods during the two severe attacks with an increase in the percentage of attackers to 50% is shown in Figure 2c. It

| | MNIST | | Fashion-MNIST | | CIFAR10 | | FEMNIST | |
|---|---|---|---|---|---|---|---|---|
| | FedAvg | FedQV | FedAvg | FedQV | FedAvg | FedQV | FedAvg | FedQV |
| Data Poison | | | | | | | | |
| Labelflip | 98.81±0.03 | 98.54±0.05 | **86.70±0.02** | 85.22±0.05 | 66.88±0.48 | 67.36±0.22 | 74.92±2.55 | **78.42±0.65** |
| Gaussian | 9.68±0.41 | 10.49±0.46 | 10.00±0.00 | **27.38±17.38** | 15.29±0.57 | **19.76±3.66** | 4.64±0.13 | 4.83±0.25 |
| Backdoor | | | | | | | | |
| ACC(%) | 37.38±19.82 | **98.30±0.15** | 74.27±9.12 | **78.40±3.95** | 59.85±2.18 | 60.65±1.72 | 49.78±22.38 | **75.20±3.96** |
| ASR(%) | 68.49±22.00 | **0.19±0.07** | 14.58±12.53 | **7.05±6.35** | 18.20±5.27 | **3.21±1.30** | 30.88±7.52 | 28.26±9.57 |
| Scaling | | | | | | | | |
| ACC(%) | 10.33±0.05 | 11.16±0.88 | 10.22±0.09 | 11.27±0.99 | 10.00±0.00 | **28.55±18.55** | 26.30±21.55 | **64.80±1.38** |
| ASR(%) | 99.94±0.06 | 98.96±1.04 | 99.74±0.10 | 98.21±1.45 | 100.00±0.00 | 67.66±32.34 | 0.47±0.08 | 0.56±0.06 |
| Neurotoxin | | | | | | | | |
| ACC(%) | 81.17±15.39 | **95.73±1.45** | 70.00±7.85 | **79.58±1.60** | 22.40±7.16 | **45.40±3.22** | 47.29±18.07 | **79.99±0.70** |
| ASR(%) | 23.19±2.25 | **18.11±1.67** | 20.65±2.21 | 18.12±4.16 | 51.63±1.03 | 57.42±1.91 | 40.42±4.35 | **9.00±1.29** |
| Model Poison | | | | | | | | |
| Krum | 10.57±0.39 | **97.96±0.14** | 10.00±0.00 | **79.43±0.86** | 10.00±0.00 | **53.27±1.12** | 5.20±0.22 | **51.86±3.06** |
| Trim | 10.04±0.16 | **98.36±0.11** | 10.00±0.00 | **84.45±0.70** | 10.00±0.00 | **57.33±2.34** | 5.09±0.33 | **52.19±4.52** |
| Min-Max | 35.00±25.38 | **85.32±6.45** | 10.00±0.00 | **67.25±7.44** | 10.00±0.00 | **19.07±6.97** | 56.37±13.67 | **72.58±2.11** |
| Min-Sum | 96.69±0.94 | 95.97±0.59 | 10.88±0.87 | **83.93±0.81** | 17.40±4.27 | **43.94±3.56** | 52.56±23.91 | **72.36±1.61** |
| QV-Adaptive | **71.43±22.67** | 56.94±23.95 | 35.92±4.60 | **62.13±11.25** | 10.00±0.00 | 11.14±1.14 | 22.08±18.72 | **43.78±20.72** |

Table 1: Comparison of FedQV and FedAvg on four benchmark Datasets under 10 attack scenarios with 30% malicious parties. The Best Results are highlighted in bold.

| | Multi-Krum | Multi-Krum + FedQV | Trimmed-Mean | Trimmed-Mean + FedQV | Rep | Rep + FedQV |
|---|---|---|---|---|---|---|
| Neurotoxin | | | | | | |
| 1% | 78.99±1.03/1.05±0.01 | 80.61±0.66/0.86±0.17 | 85.23±1.77/0.59±0.13 | 84.75±0.84/0.45±0.06 | 80.99±1.15/0.84±0.32 | **85.82±0.55/0.38±0.06** |
| 5% | 76.21±0.73/3.29±0.92 | 80.32±1.07/**1.34±0.34** | 85.15±0.38/0.87±0.17 | 85.09±0.97/0.73±0.05 | 80.48±1.17/1.62±0.11 | **84.12±0.38/1.35±0.40** |
| 10% | 72.79±1.02/21.73±7.07 | **77.41±1.36/16.30±3.01** | 85.13±0.70/2.32±0.27 | 84.68±0.73/1.45±0.21 | 80.86±0.86/1.30±0.10 | **83.78±0.09/0.66±0.04** |
| Min-Max | | | | | | |
| 10% | 71.62±4.48 | **79.48±0.82** | 75.41±0.77 | 78.46±0.67 | 72.66±1.34 | **76.98±1.48** |
| 30% | 52.29±0.34 | **58.42±4.92** | 59.62±1.20 | 60.24±6.81 | 54.94±0.82 | **58.02±0.71** |
| 50% | 10.28±0.28 | **22.95±9.94** | 9.47±0.42 | 10.64±0.63 | 11.23±1.00 | 13.64±2.58 |
| QV-Adaptive | | | | | | |
| 10% | 52.98±1.78 | **73.35±3.44** | 83.17±1.85 | **85.55±0.33** | 12.60±2.36 | **41.55±19.78** |
| 30% | 34.93±14.60 | **55.07±11.40** | 29.14±19.14 | 42.17±25.95 | 12.44±2.44 | **38.44±14.32** |
| 50% | 10.20±0.20 | 12.24±1.12 | 10.00±0.00 | **13.35±3.37** | 10.00±0.00 | 10.55±0.44 |

Table 2: Comparison of Multi-Krum, Trimmed-Mean, Reputation, and their integration with FedQV under SOTA attacks. The best results are in bold. The results under targeted attacks are "ACC / ASR".

demonstrates that the combination of FedQV and the reputation model considerably strengthens resistance against Gaussian and Scaling attacks by at least a factor of 26%. Setting this observation as the alternative hypothesis $H_1$ and using the Wilcoxon signed-rank test, we can reject the null hypothesis $H_0$ at a confidence level of 1% in favour of $H_1$.

## 5.7 INTEGRATION WITH BYZANTINE-ROBUST AGGREGATION

Our objective is not to position FedQV in competition with existing defence techniques but rather to demonstrate that FedQV can act as a complementary approach to advanced defences. FedQV can be seamlessly integrated into Byzantine-robust defence by adapting the weight calculation process. We illustrate this with examples using Muilt-Krum Blanchard et al. (2017), Trim-mean Yin et al. (2018) and Reputation Chu et al. (2022). Figure 2d shows that the accuracy of Multi-Krum increases considerably, especially for local poisoning attacks. Table 5 demonstrates that the integration of the Multi-Krum, Trimmed-Mean and Reputation method with FedQV leads to superior performance(Higher ACC and lower ASR) compared to the standalone versions. These findings support that FedQV holds promise as a valuable complementary method to existing defence mechanisms.

## 6 CONCLUSION

In this paper, we have proposed FedQV, a novel aggregation scheme for FL based on quadratic voting instead of *1p1v*, which is the underlying principle that makes the currently employed FedAvg vulnerable to poisoning attacks. The proposed method aggregates global models based upon the votes from a truthful mechanism employed in FedQV. The efficiency of the proposed method has been comprehensively analysed from both a theoretical and an experimental point of view. Collectively, our performance evaluation has shown that FedQV achieves superior performance than FedAvg in defending against various poisoning attacks. Moreover, FedQV is a reusable module that can be used in conjunction with reputation models to assign unequal voting budgets, other Byzantine-robust techniques, and privacy-preserving mechanisms to provide resistance to both poisoning and privacy attacks. These findings position FedQV as a promising complement to existing aggregation in FL.

