# OpenReview forum: "FedQV: Leveraging Quadratic Voting in Federated Learning"
_ICLR.cc/2024/Conference — Submitted to ICLR 2024_

### Official Review · Reviewer_E32F · 2023-10-15

**Soundness:** 3 good
**Presentation:** 3 good
**Contribution:** 2 fair
**Rating:** 3
**Confidence:** 3

**Summary:**

The authors propose a new aggregation scheme for FL instead of regular FedAvg. For this, they draw on quadratic voting proposed in the election literature and design a scheme where the weighting of submitted gradients is determined based on both the similarity to the previous global model and a voting budget. The authors also briefly discuss an optimization/extension that includes a reputation system. Notably, the scheme is compatible with existing privacy-preserving and robust aggregation schemes. The authors successfully evaluate the resistance of their scheme when training for standard image classification tasks under a wide range of targeted as well as untargeted data and model poisoning attacks.

**Strengths:**

The paper is well-written and accessible for non-experts.

The suggested scheme is compatible with secure and robust aggregation schemes.

The claims on convergence and truthfulness are theoretically proven.

The authors evaluate the effectiveness on a wide range of both untargeted and targeted attacks instead of focusing on only a specific type of attack.

**Weaknesses:**

References and appendix are not included in the submission PDF and can only be found in the supplementary material.

It is unclear why the similarity computation takes place on the client instead of server side. Moreover, it is unclear why the server simply relies on the submitted similarity score s instead of verifying this computation. I assume this is for compatibility with secure aggregation methods where the server does not see gradients in the clear. However, this is not discussed and gives opportunity for trivial attacks where the client submits an extremely harmful w but lies about similarity s. Likewise, it might be worth to discuss what happens if advanced attackers optimize their malicious w to cause maximum damage while still keeping the similarity just below the detection threshold.

It is furthermore unclear to me how the proposed "masked" voting rule is supposed to prevent the client from tracking their voting budget. The propsed rule H simply is a log calculation over the normalised similarity s. Intuitively, the clients can thus at least roughly estimate its budget and act accordingly.

The paper assumes a very high number of malicious clients in the system, e.g., it discusses cases where "attackers manage to control the majority of votes, then via poisoning their tyranny will manifest itself as a degradation of the accuracy of the FL model used by the minority". In the experiments, between 10%-50% is assumed. However, Shejwalkar et al. (SP'22) have shown that in cross-device production systems the corruption is below 0.1%. It remains unclear how FedQV performs in these settings.

The considered data poisoning attacks are not all state of the art. For example, the authors consider simple static label flipping (Fang et al.) but not more advanced dynamic label flipping attacks (Shejwalkar et al., SP'22).

The paper states that robust aggregations cannot be combined with secure aggregation. However, works like HyFL (arXiv:2302.09904) implement such aggregation schemes in secure computation.

**Questions:**

- Why does the server not compute and verify the similarity measure?
- How does the masked voting rule prevent the client from roughly tracking/estimating the remaining budget?

---

> ### Author Response · Authors · 2023-11-16
> **Thanks for the reviews!**
>
> Thank you for your thoughtful and insightful feedback on our paper. We reply to the questions raised one by one:
>
> **A1**: The decision to calculate the similarity score on the client side is driven by the need to prevent the server's access to gradients, mitigating privacy risks such as model inversion attacks. It also aligns with compatibility requirements for secure aggregation methods.
>
> We acknowledge the potential vulnerability that the reviewer pointed out. Hence in our experimental evaluation, we have already evaluated a range of such attack scenarios:
>
> (i) Data poisoning attacks: Attackers submit the similarity score based on their poisoned updates;
>
> (ii) Model poisoning attacks: Attackers submit the true similarity score based on their clean updates and poison their model updates;
>
> (iii) Adaptive attack: Attackers manipulate the similarity score and submit poisoned updates.
>
> The collective results from these attacks are that FedQV is more robust than FedAVG in all of them, as shown in Table 1.
>
> **A2:** The client's inability to track or estimate the remaining budget is ensured through two key mechanisms. First, the vote calculation is performed exclusively on the server side, as indicated in Algorithm 1 Lines 6 to 8. Clients lack access to information about their budget; only the server possesses this knowledge. Second, even if clients were aware of the voting rule, their lack of knowledge about the normalized similarity score, which depends on others' similarities, prevents them from determining their votes. This design reinforces the privacy and security aspects of the proposed masked voting rule.
>
>
> **A3 (Percentage of corruption):** The reviewer’s point about the percentage of attackers is well-taken. The choice of this percentage is guided by the consideration of potential scenarios involving a larger number of attackers in which the damage to the global model can be very substantial. However, we completely acknowledge the importance of evaluating FedQV also under more realistic lower attack percentages. We have already conducted evaluations with low percentages such as 1%, 5%, and 10%, but placed these results in Appendix Table 6 due to the page limit. This supplementary result illustrates that FedQV continues to perform effectively even with lower attacker percentages. We will make a more clear note on this in the revised version.
>
>
> **A4 (Advanced version of label flipping attack)**: We appreciate the observation regarding the advanced label flipping attacks and acknowledge the existence of more advanced attacks. In our evaluation, we tested FedQV against a total of 9 different poisoning attacks, including 3 state-of-the-art attacks published last year. The rest are classic poisoning attacks, which are included for comparability with existing byzantine-robust defences. While we believe these attacks are sufficient to support our findings, recognizing the importance of the attack mentioned by the reviewer, we are open to incorporating it in the camera-ready version to further enhance the comprehensiveness of our evaluation.
>
> **A5 (HyFL)**: In our paper, we state that robust aggregations are difficult to combine with secure aggregation due to the heavy computation, which is also mentioned In HyFL. In the HyFL framework, a Hierarchical FL approach incorporates a lightweight variant of the Trimmed Mean defence designed for fighting against data-poisoning attacks. In this context, FedQV can be implemented atop the Trimmed Mean variant, serving as an additional layer of defence against data poisoning attacks.

---

> > ### Comment · Reviewer_E32F · 2023-11-29
> > **Response to Rebuttal**
> >
> > Thanks for the comments and clarifications! Re A1: It still appears to me that the fact that the server accepts the submitted similarity scores unchecked is a fatal flaw of the paper. I'm not sure the listed experiments are sufficient to assess the impact of this vulnerability. For example, running data poisoning attacks while submitting wrong similarity scores will go completely unnoticed by the scheme. The adaptive attack is not clearly described in the main body of the paper, so I'm not sure this captures the mentioned attack scenario.

---

> > > ### Author Response · Authors · 2023-11-29
> > > **Thanks for the feedbacks!**
> > >
> > > Thank you for your continued engagement with our work. We'd like to provide additional insights into the robustness of FedQV against potential vulnerabilities.
> > >
> > > Firstly, regarding the client-side calculation of the similarity score, we respectfully disagree with the characterization of a fatal flaw for several reasons:
> > > - The server does examine the received similarity scores from clients by using a masked voting rule, as outlined in Algorithm 1. This process efficiently filters out any abnormal or suspicious values.
> > > - Even in instances where a manipulated similarity score succeeds in being accepted, the inherent constraints of QV act as a limiting factor, restricting the influence it can exert and thereby minimizing potential damage.
> > > - The employed mechanism also serves as a defense against privacy attacks, impeding the server from engaging in privacy-threatening calculations and reinforcing the overall security of the system.
> > >
> > > Secondly, the true strength of FedQV lies not in being a standalone defense but in its role as a complementary element to existing FL defense strategies. As demonstrated in Section 5.7, integrating Byzantine-robust aggregation methods with FedQV significantly enhances robustness against various poisoning attacks. This underscores FedQV's suitability as a robust foundation for such defense methods, showcasing its collaborative effectiveness in bolstering FL security.
> > >
> > > Lastly, although due to page limitations, we have placed the details of the adaptive attack in the appendix C.3, this attack demonstrates the manipulation of both similarity scores and model updates as you suggested. The procedure involves:
> > > 1. The malicious client $i$ generates benign updates $w_{i}^{t}$ using unpoisoned data $D_{i}$ in round $t$ and calculates the corresponding similarity score.
> > > 2. Malicious clients (with counts of $m$) collectively normalize all the similarity scores. They optimize the similarity score using AGR-agnostic Min-Max to manipulate the scores of malicious clients, enhancing the likelihood that the resulting scores are accepted by the server.
> > > 3. Then the attackers optimize the local model poisoning by optimizing the following problem:
> > > $$\max_{ \upsilon} \upsilon$$
> > >
> > > $$\text{s.t.} {w^{t}}^{'}_{i \in m} = \text{FedQV}(w_1^{t},w_2^{t},\cdots,w_m^{t})$$
> > >
> > > $${w^{t}}^{'}_{i \in m} = w^{t} - \upsilon \hat{d}$$
> > >
> > > Here, $\hat{d}$ represents a column vector encompassing the estimated changing directions of all global model parameters. The variables $w_{i \in m}^{t}$ and ${w^{t}}^{'}_{i \in m}$ correspond to the local model $i$ before and after the attack, respectively. The parameter $\upsilon$ denotes the extent of the attack's impact on the model.

---

### Official Review · Reviewer_dir9 · 2023-10-30

**Soundness:** 2 fair
**Presentation:** 2 fair
**Contribution:** 2 fair
**Rating:** 3
**Confidence:** 3

**Summary:**

Federated learning is vulnerable to poisoning attacks by malicious participants. This gets aggravated when the influence of parties is proportional to the amount of data they have. This is due to the fact that malicious participants that claim to have large datasets have a big impact when they send corrupted updates.

This work presents FedQV, a technique to improve resilience to byzantine attacks in federated learning by leveraging ideas from different domains. First, the work takes ideas from quadratic voting, a technique for robust aggregation of weighted contributions that reduces the otherwise-large influence of participants that possess big datasets. Second, it proposes a control mechanism that requires users to publish the similarity of their updates with respect to the previous global model. Third, it proposes a truthful mechanism that incentives parties to upload consistent contributions to maximize its own utility. Finally, it provides a reputation based approach which assigns different weights to contributions depending on their past faults.

The paper theoretically shows convergence in the presence of malicious adversaries as well as improved resilience when compared to regular Federated Averaging. In addition, it shows improved resilience when FedQV is combined with other defense measures taken by the server.

**Strengths:**

The paper proposes novel ideas by taking techniques from different domains.

I think that the inclusion of incentive-based approaches to deal with corrupted parties is currently less explored that other techniques and can potentially provide new means to understand which attacks are realistic. Therefore, this is a valuable aspect of the contribution.

Other ideas such as the integration of robust voting techniques and the use client-side measures are original and constructive in the discussion of byzantine resilience. In particular, techniques that are compatible with privacy enhancing technologies such as secure aggregation are valuable, especially given that an important use case of federated learning is privacy preserving machine learning.

**Weaknesses:**

However, the contribution presents major problems which are listed below:

1- The threat model of the paper is not clear. A well defined threat model is crucial when studying defenses against active adversaries. In the current contribution, I do not see what are the adversaries' possible behaviors and what are they willing to risk to harm the model (e.g. are they able to collude?). Therefore, it is not clear in practice in which scenarios would the truthfulness and convergence properties hold. This makes hard to evaluate the real impact of the contribution.

For example, consider an adversary which controls many parties and each party sends a harmful update in a different round. The adversary is willing to sacrifice parties to harm the model. Would this affect the truthfulness property?

2- Clarity. The contribution is a combination of many ideas taken from different domains. However, the inclusion of each ingredient is not properly justified:

2a- The paper introduces quadratic voting (QV) as a defense to reduce the impact of malicious users that falsely claim to have large datasets. However, the paper does not show how this kind of attack can affect other current Federated Learning defenses in practice and how QV is an effective defense. Moreover, the paper does not evaluate the resilience against any attack in which the adversary falsely claims to have a large dataset. Therefore, I am not sure if the initial motivation of QV is present in practice.

2b- The paper introduces adaptive budgets as a reputation system. The original FedQV (Algorithm 1) already seems to penalize contributions. Therefore, the reputation protocol (Algorithm 2) appears a bit redundant. In the following sections of the paper, it is not clear when FedQV-Alg1 and FedQV-Alg2 are applied.

2c- Compatibility with privacy enhancing primitives. In Section 3.2, the paper claims that the compatibility with privacy enhancing primitives (Secure Aggregation, Fully Homomorphic Encryption, Differential Privacy) is one of its main benefits of FedQV. However, I do not see a clear compatibility. For example, in Secure Aggregation the aggregation is not done by the server. Therefore, it not clear (i) how can the server adjust the weights of the aggregation properly and (ii) how can clients share similarity scores and local dataset size with the server without breaking the privacy guarantees.

2d- Meaning of the truthfulness property: Section 4.2 claims that any possible corrupted contribution diminishes the probability of harm as server penalties reduce the influence of corrupted parties. This gives the impression that being honest is better for local utility than performing an attack. However, this is not what we can see empirically in section 5.

3- Evaluation of FedQV:  the proposed protocol claims to be easy to integrate with other resilience measures because similarities are computed in the client side. As said, this is an interesting feature, but is particularly vulnerable to client-side manipulations of the similarity scores and dataset sizes. It is true that protocol is evaluated against QV-adaptive, an attack that exploits this vulnerability. However, delegating the control of the defenses to possibly malicious clients is a high risk to take. Therefore, its evaluation of this risk requires a more comprehensive treatment.

For example, an important aspect of the evaluation which I feel is missed is (as said in point 2a) the resilience against an attack in which the adversary falsely claims to have a large dataset.

**Questions:**

Please elaborate on the weaknesses outlined above.

---

> ### Author Response · Authors · 2023-11-16
> **Thanks for the reviews!**
>
> Thank you for your thoughtful and insightful feedback on our paper. We reply to the questions raised one by one:
>
> 1- The reviewer is correct that the threat model is crucial. In our paper, attackers can collude based on the type of attack. For instance, for example, the QV-adaptive attack in Section 5, where collusion is used to manipulate similarity scores. The results in Table 2 demonstrate that FedQV effectively defends against this collusion-based attack.
>
> The truthfulness property is considered from each party's perspective, focusing on individual utility. This implies that malicious parties also aim to remain in the system while contributing to model harm. Sacrificing some colluders wouldn't yield better results than our considered scenario because our designed QV-adaptive attack avoids such sacrifices, ensuring more powerful harm without compromising colluders unnecessarily. However, even if the manipulated similarity scores are accepted, the inherent constraints of QV, as shown in Figure 1, limit the influence the attackers can exert through restricted aggregation weights, minimizing their harm to the global model. This aspect will be clarified in the revised version for better understanding.
>
> 2.
>
> 2a-  We acknowledge the risk of malicious users falsely claiming larger datasets and introduce Quadratic Voting (QV) as a defence, specifically evaluating its impact on FedAvg. Figure 1 illustrates how the attack works when a malicious party claims to possess a larger dataset size. The result shows that having or claiming to have a larger dataset does not succeed in harming the model under FedQV.
> Furthermore, it is crucial to note that many current FL Byzantine-robust defences, such as Krum, trimmed-mean, and Rep, do not rely on the size of the dataset as the sole determinant for aggregation weights. Unlike FedAvg, where the attack in question could be effective, these defences utilize alternative aggregation mechanisms that are not susceptible to the same type of manipulation.
>
> 2b- FedQV-Alg2's introduction is not redundant; it strategically assigns uneven budgets, rewarding benign peers and limiting malicious influence. This boosts performance by at least 26%, as shown in Fig. 2c compared to the baseline FedQV.
> In the sections preceding 5.6, we assess the outcomes using the naive FedQV. Subsequently, in Section 5.6, we conduct evaluations of FedQV with reputation (denoted as FedQV + REP) in Figure 2c. The revised version will offer additional clarity on this distinction.
>
> 2c- Regarding the compatibility with privacy-enhancing primitives:
>
> (i) To achieve secure aggregation in FedQV, clients initiate the process by sharing their similarity scores with the server. The server, utilizing the masked voting rule, adjusts the weights and broadcasts them back to the clients. Subsequently, clients upload gradients, weights, and local dataset sizes to the aggregator in a secure aggregation process;
>
> (ii) Firstly, the server remains unaware of the local dataset sizes of individual clients. Secondly, the server only possesses similarity scores, making it impossible to reconstruct the original gradients of clients. This safeguards against potential privacy breaches.
> Implementing the confidential computation on top of QV is part of our ongoing work, and we are actively implementing and refining these privacy-preserving mechanisms.
>
> 2d- It's important to clarify that local utility for clients is rooted in enhancing their influence on the model, aiming for more aggregation weights and increased participation in FL training rounds. From this perspective, being honest aligns with their goal. However, as attackers seek to harm the model, they may strategically attempt to maximize their harm while remaining within the system. This behaviour is evident in Section 5. We are happy to provide additional clarification in the revised version to ensure a comprehensive understanding.
>
> 3- We acknowledge the potential vulnerability that the reviewer pointed out. Hence in our experimental evaluation, we have already considered a range of attack scenarios:
> - Data poisoning attacks: Attackers submit the similarity score based on their poisoned updates.
> - Model poisoning attacks: Attackers submit the true similarity score based on their clean updates and poison their model updates.
> - Adaptive attack: Attackers manipulate the similarity score and submit poisoned updates.
>
> Regarding the attack where adversaries falsely claim to have a large dataset, this attack can be successful against FedAvg (as mentioned in point 2a). But against FedQV it cannot succeed because, as illustrated in Figure 1, FedQV effectively excludes the malicious party from the aggregation process, thereby improving the accuracy of the resulting global model. This demonstrates FedQV's efficacy as a defence against this attack compared to FedAvg.

---

> > ### Comment · Reviewer_dir9 · 2023-12-01
> > **Reply to the rebuttal**
> >
> > Dear authors,
> >
> > I have read your rebuttal and appreciate your response. However,
> >
> > - I still think the technique requires a clearer assessment of its vulnerabilities and a better explanation of the choice of each element of the construction as part of the defense (i.e. why is element X helping in the defense of attack Y?)
> >
> > - I don't agree with the authors in the compatibility with privacy enhancing primitives (point 2c). Sharing external information in addition to the output of the primitive (e..g similarity scores) would probably break the privacy that the primitive is providing.
> >
> > Therefore, my score remains unchanged.

---

### Official Review · Reviewer_WYWh · 2023-11-23

**Soundness:** 2 fair
**Presentation:** 2 fair
**Contribution:** 2 fair
**Rating:** 5
**Confidence:** 3

**Summary:**

The paper introduced the quadratic voting scheme into federated learning to replace the traditional 1p1v strategy. The idea of allowing the participation parties to decide their participation weight in each round is interesting but I am not sure how useful it is to prevent Byzatine attacks. The authors also integrate other countermeasures to make the scheme Byzantine-robust like suppressing gradients different from the majority and allocating budget based on reputation.

**Strengths:**

The idea of allowing FL participants to decide how to participate in FL using budget is novel. However, I am not sure how useful it is to prevent against Byzantine attackers since this gives the attacker more attack space to explore. For instance, comparing to limiting each client to participate in at most k rounds, the QV strategy of giving a client budget k is a super set and gives the adversary more freedom to design their attacks.

**Weaknesses:**

In the evaluation section, FedQV is only compared with FedAvg. There are many existing Byzantine-robust FL strategies that should be compared with
[1] Attack-resistant federated learning with residual-based reweighting. Shuhao Fu, Chulin Xie, Bo Li, and Qifeng Chen.
[2] Robust aggregation for federated learning. Krishna Pillutla, Sham M Kakade, and Zaid Harchaoui.
[3] Byzantine stochastic gradient descent. Dan Alistarh, Zeyuan Allen-Zhu, and Jerry Li.
[4] Learning from history for byzantine robust optimization. Sai Praneeth Karimireddy, Lie He, and Martin Jaggi.
[5] Byzantine-Robust Federated Learning with Optimal Rates and Privacy Guarantee. Banghua Zhu, Lun Wang, Qi Pang, Shuai Wang, Jiantao Jiao, Dawn Song, Michael I.Jordan.
[6] Byzantine-resilient non-convex stochastic gradient descent. Zeyuan Allen-Zhu, Faeze Ebrahimian, Jerry Li, and Dan Alistarh.
[7] Byzantine-robust learning on heterogeneous datasets via bucketing. Sai Praneeth Karimireddy, Lie He, and Martin Jaggi.
[8] Variance reduction is an antidote to byzantines: Better rates, weaker assumptions and communication compression as a cherry on the top. Eduard Gorbunov, Samuel Horváth, Peter Richtárik, and Gauthier Gidel.
[9] Secure byzantine-robust distributed learning via clustering. Raj Kiriti Velicheti, Derek Xia, and Oluwasanmi Koyejo.
[10] Byzantine-resilient high-dimensional sgd with local iterations on heterogeneous data. Deepesh Data and Suhas Diggavi.

**Questions:**

Please compare with other existing Byzantine-robust FL systems.

---

> ### Author Response · Authors · 2023-11-24
> **Thanks for the reviews!**
>
> Thank you for your valuable feedback on our paper. Regarding the question raised, it's crucial to note that our objective is not to position FedQV in competition with existing Byzantine-robust FL strategies but rather to demonstrate that FedQV can act as a complementary approach to advanced strategies. Through the integration of FedQV with other state-of-the-art defence strategies, our aim is to establish a comprehensive defence framework capable of effectively tackling a broader spectrum of challenges within FL. As showcased in Sections 5. 6 and 5.7, our experiments highlight how FedQV can seamlessly join forces with other defence methods, such as  Muilt-Krum Blanchard et al. (2017), Trim-mean Yin et al.(2018) and Reputation Chu et al. (2022)., to achieve heightened performance collaboratively, rather than engaging in direct competition.

---

### Official Review · Reviewer_vHfa · 2023-11-30

**Soundness:** 3 good
**Presentation:** 3 good
**Contribution:** 2 fair
**Rating:** 5
**Confidence:** 4

**Summary:**

The paper considers the federated learning setting, and investigates new aggregation strategies for FedAvg that make it more robust to malicious users. The method falls within the framework of voting-based aggregation strategies and is inspired by quadratic voting. The authors analyze their algorithm (FedQV), showing it has similar convergence guarantees to FedAvg, and that it is a truthful mechanism (roughly the dominant strategy of each user is to play truthfully). The paper provides empirical results comparing FedQV to FedAvg, showing that the new method has better robustness to existing attacks (and some new attacks tailored for FedQV). Both methods have similar performance on the training set, however, FedAvg enjoys better performance on the test set.

**Strengths:**

The paper is well-written and the definitions\results are clearly stated. The experiments seem promising, showing better robustness to many of the existing attacks. However, I have some concerns as I write below.

**Weaknesses:**

Perhaps the main downside of this work is missing a necessary comparison to other existing algorithms rather than FedAvg, especially ones which were designed to be more robust. The authors could also compare against more robust versions of FedAvg where the aggregation limits the norm of each user update.

It is therefore hard to evaluate the significance of the new method without being able to see its improvements over existing methods (or standard robustifications of FedAvg).

Another limitation of FedQV is that its accuracy on the test set is worse than FedAvg (with a big gap for some datasets).

Based on the above, I recommend to delay the acceptance of the paper to a future conference where these issues could be improved.

More comments:

1. It seems that in many case the reputation strategy was able to improve the robustness significantly. Thus, it would be interesting to compare FedAvg + Rep against FedQV + Rep to understand if the main improvements are due to quadratic voting or the reputation strategy.

2. The paper has to have clear definitions for the notation, e.g. in the definition of L(w), the authors need to clearly define \mc{X}.

3. Plots (a) and (b) in Figure 1 have too many details (and datasets). I’d suggest to separate the plots and have a different plot for each dataset, to make it easier to understand the figures.

4. The presentation is good overall but could be improved in certain parts of the paper: for example, assumptions are only presented in appendix and so the notation in theorem 1 is not clear without going to the appendix. Moreover, the section on truthfulness (section 4.2) lacks some more intuitions and explanations, e.g. 1. Why would the user want to maximize v_i(a) - p_i(v_i,v)? and 2. What is v_i(a) in definition 4.4?.

**Questions:**

See above.

---

> ### Author Response · Authors · 2023-12-01
> **Thanks for the reviews!**
>
> Thank you for your thoughtful and insightful feedback on our paper. We reply to the questions raised one by one:
>
> **A1. comparison with other existing algorithms**: It is crucial to note that our objective is not to position FedQV in competition with existing Byzantine-robust FL algorithms but rather to demonstrate that FedQV can act as a complementary approach to advanced algorithms. Through the integration of FedQV with other state-of-the-art defence strategies, we establish a comprehensive defence framework capable of effectively tackling a broader spectrum of challenges within FL. As showcased in Sections 5. 6 and 5.7, our experiments highlight how FedQV can seamlessly join forces with other defence methods, such as Muilt-Krum Blanchard et al. (2017), Trim-mean Yin et al.(2018) and Reputation Chu et al. (2022)., to achieve heightened performance collaboratively, rather than engaging in direct competition.
>
> **A2. Comparison with more robust versions of FedAvg**: In Section 3.1, when discussing the benefits of FedQV, we highlight its compatibility with various FL mechanisms. Notably, techniques like clipping and nor, which contribute to the robustness of FedAvg, can seamlessly integrate into FedQV. Therefore, the primary focus remains on comparing FedQV with FedAvg, considering that additional modules designed for FedAvg can also be applied to FedQV.
>
> **A3. Accuracy of FedQV**: In the comparison of accuracy between FedQV and FedAvg, as depicted in Fig 1(a), FedQV demonstrates comparable accuracy to FedAvg in the absence of attacks. However, under attack scenarios, as evidenced by the results presented in Table 1 and Fig 1(b), FedQV exhibits significantly improved accuracy compared to FedAvg.
>
> **A4. Reputation Scheme**: To assess the contribution of quadratic voting (QV) versus the reputation scheme, we compare FedQV + Rep with Rep in Section 5.7. The results indicate that incorporating FedQV enhances the robustness of the reputation scheme, underscoring the effectiveness of FedQV as a complementary element to existing FL defense algorithms.
>
> **A5. Notation and Presentation Clarity**:
> - In the definition of L(w), \mathcal{X} represents the Non-IID distribution where the dataset is drawn from, as described in Section 3.1.
>
> - Regarding Figure 1: We appreciate the suggestion to enhance clarity. We will revise the plots, separating them for each dataset to improve understanding.
>
> - Presentation Clarity: Due to page limitations, assumptions are currently in the appendix. In the revised version, we will present key assumptions in the main body of the paper to improve the clarity of notation in theorems and throughout the document.
>
> **A6. Truthfulness**
> - The expression  v_i(a) - p_i(v_i,v_-i) represents the utility of party $i$, which means the gain from voting (v_i(a)) minus its cost (p_i(v_i,v_-i) ). This formulation ensures that telling the truth is more beneficial than lying(v_i(a) - p_i(v_i,v_-i) > v_i(a') - p_i(v'_i,v_-i).
>
> -  v_i(a) in Definition 4.4 denotes the gain of party $i$ if the outcome of the voting is a. In the revised version, we will provide more intuition and explanation to enhance clarity.

---

### Meta-Review · Area_Chair_PiYh · 2023-12-14

**Metareview:**

This paper proposes and evaluates an update strategy in FL based on quadratic voting scheme. This scheme downweighs the contributions from users whose update are more dissimilar from the current model and uses a budget to further limit the total effect of the user. The proposed scheme is shown to be more robust to a specific poisoning attack. While the basic problem of ensuring robustness to poisoning is important, the work does adequately compare the the proposed technique with other simple approaches to increasing robustness. This is especially problematic since the technique reduces the quality of the model on a number of datasets. One notable omission is the lack of comparison with techniques developed in the context of differential privacy requires limiting the sensitivity of the resulting model to individual user's data and uses technique such as clipping of contributions and individual budget tracking across updates.

**Justification For Why Not Higher Score:**

n/a

**Justification For Why Not Lower Score:**

n/a

---

### Decision · Program_Chairs · 2024-01-16

Reject